# The effect of emotional information from eyes on empathy for pain: A subliminal ERP study

Juan Song[1]*, Yanqiu Wei[1], Han Ke[2]

**1** Key Research Base of Humanities and Social Sciences of the Ministry of Education, Academy of Psychology and Behavior, Faculty of Psychology, Tianjin Normal University, Tianjin, China, **2** Psychology, School of Social Science, Nanyang Technological University, Singapore

* songjuan@tjnu.edu.cn

**Data Availability Statement:** All data underlying our findings described in the manuscript can be fully available without restriction within the manuscript. Figures were the supporting information for our research. Our original data is available upon request (including the ERP

## Abstract

Facial expressions are deeply tied to empathy, which plays an important role during social communication. The eye region is effective at conveying facial expressions, especially fear and sadness emotions. Further, it was proved that subliminal stimuli could impact human behavior. This research aimed to explore the effect of subliminal sad, fearful and neutral emotions conveyed by the eye region on a viewer's empathy for pain using event-related potentials (ERP). The experiment used an emotional priming paradigm of 3 (prime: subliminal neutral, sad, fear eye region information) × 2 (target: painful, nonpainful pictures) within-subject design. Participants were told to judge whether the targets were in pain or not. Results showed that the subliminal sad eye stimulus elicited a larger P2 amplitude than the subliminal fearful eye stimulus when assessing pain. For P3 and late positive component (LPC), the amplitude elicited by the painful pictures was larger than the amplitude elicited by the nonpainful pictures. The behavioral results demonstrated that people reacted to targets depicting pain more slowly after the sad emotion priming. Moreover, the subjective ratings of Personal Distress (PD) (one of the dimensions in Chinese version of Interpersonal Reactivity Index scale) predicted the pain effect in empathic neural responses in the N1 and N2 time window. The current study showed that subliminal eye emotion affected the viewer's empathy for pain. Compared with the subliminal fearful eye stimulus, the subliminal sad eye stimulus had a greater impact on empathy for pain. The perceptual level of pain was deeper in the late controlled processing stage.

## Introduction

Empathy refers to the process of an individual sharing and understanding another person's emotions, feelings and thoughts. It plays an important role in communication and interaction in human society [1]. Gladstein proposed two major types of empathy: affective empathy (responding with the same emotion to another person's emotion) and cognitive empathy (intellectually taking the role or perspective of another person) [2]. Further, Davis divided empathy into 4 parts: perspective-taking (tendency to spontaneously adopt the psychological point of view of others), empathy concern ("other-oriented" feelings of sympathy and concern

waveforms, AVG and original data). All the means and SD are included in the manuscript. Our data did not extract any points from images. The images are the reflection of our data. The ERP waveforms' figures' making were based on the mean data of our subjects. If it is really needed, we can supply our grand mean AVG file for ERP waveforms.

**Funding:** The research in the manuscript was funded by Tianjin Philosophy and Social Science Projects (China) Number: TJJX17-010 to JS. The funder had no role in study design, data collection and analysis, decision to publish, or preparation of the manuscript.

**Competing interests:** The authors have declared that no competing interests exist.

for unfortunate others), fantasy (tendency to transpose themselves imaginatively into the feelings and actions of fictitious characters in books, movies, and plays) and personal distress ("self-oriented" feelings of personal anxiety and unease in tense interpersonal settings) [3]. Modern accounts of empathy decompose empathy into these three components: experience-sharing involves vicariously resonating with others' experiences, mentalizing involves actively inferring others' thoughts and intentions, and compassion involves motivation to alleviate others' suffering [4, 5]. Empathy is regarded as a suitable explanation for prosocial behavior [6], such as taking care of offspring and cooperating with others.

Empathy for pain has been the major focus of research devoted to empathy in social neuroscience and other related fields, making it the most dominant neuroscientific domain in the study of empathy [7]. A review paper summarized research in this field for the past twenty years, it was found that the theme of empathy research gradually shifted from early personality traits, attitudes, and emotions to social cognition [8]. With the development of technology, research on the cognitive neuroscience of empathy became a hot topic. On one hand, some researchers thought that the medial prefrontal cortex (mPFC), superior temporal sulcus (STS), temporal poles (TP), and ventromedial cortex (VM cortex) were involved in cognitive empathy, and the amygdala, insula and inferior frontal gyrus (IFG) were involved in affective empathy. On the other hand, the right temporoparietal junction (rTPJ), the insula and the anterior midcingulate cortex (aMCC) were thought to be related to empathy for pain [9]. The aMCC and the insula were the "shared neural circuit" of empathy for pain [10]. In other research, the prefrontal cortex (PFC) and the anterior cingulate cortex (ACC) were closely related to empathy for pain [11]. Some new research found that the fusiform face area (FFA) was involved in empathy tasks [12]. The mirror neuron, "neural bridge" in social communication, occupies an important place in empathy [13].

As we know, the face can provide important social information and reflect potential changes in the environment [14, 15]. People can infer and understand others' inner state immediately by observing facial expressions. A study found that individuals with high empathy paid attention to various negative facial expressions (angry and afraid faces) more than those with low empathy from very-early stage (reflected in the N170) to late-stage (reflected in the LPP) processing of faces [16]. N170 is a negative component elicited by face's feature and is sensitive to eye information. For example, isolated eye regions elicit N170 amplitudes and delayed latencies compared to faces. Hence, there is a close relationship between empathy and facial expression.

Moreover, there is special processing of eyes in facial perception. For instance, research found that the adults had obvious attentional bias for the eyes [17] using an eye movement technique. When recognizing sad emotions, the eye cues had greater significance than other cues. People first notice the eyebrow area, and the eyebrow area captured more attentional resources than the mouth area [18]. In addition, threat information conveyed by eyes was the same as that conveyed by the face when distinguishing facial expressions, but the effect of the mouth area was quite different from that of the whole face [19]. Qiao drew a similar conclusion [20]. The eye was the most expressive part of the face, especially for fearful and sad emotions. The eyes have an advantage in facial processing, and eye region information plays an important role in people's processing of negative emotion. These conclusions provided strong evidence that sad and fearful information from eye region can accurately represent facial expression.

Therefore, eye region emotional information, as a unique part of facial expression, is inseparable from empathy for pain. People can infer others' current mental state just through observing the eyebrow area of the face, and adjust their social behaviors accordingly. Empathy can also be measured by the "Reading the Mind in the Eyes" test [21], which was created by

Baron-Cohen [22]. The test reflects the ability to interpret people's complex emotional states in some situations when other information is limited. Furthermore, one study found that participants made more cooperative decisions after supraliminal eye contact than they did with no eye contact, and the effect only existed for the prosocial but not for the pro-self [23]. To date, research using eye stimuli alone to study face processing is still in an initial stage [24]. Additionally, emotions conveyed by eye region information become more important and a necessary way for people to read others' mind when expressions of the others' whole face is not available, such as in medical situations or hijacking. The study of eye emotions has many practical applications.

In summary, the eye region was found to be the important area for conveying facial expression [25]. In previous research, the ERP research concerning empathy for pain mostly used pictures and videos describing hands or feet in painful situations, or they used a painful facial expression to explore the subjects' index of brain activity and behavioral changes when observing others' pain [26, 27, 28, 29, 30]. To date, there have been a few ERP studies comparing the effect of negative emotions on empathy for pain [26, 27, 31]. For example, one supraliminal ERP study looking at the effect of eye emotional information on empathy for pain found a significant difference on P3 component induced by different eye conditions. The amplitude in neutral eyes condition was greater than that in sad eyes condition and fear eyes condition. Different supraliminal emotional information expressed by eye region affects late processing of empathy for pain [31]. Further, negative emotions have a profound influence on human survival and are important for social adaption from an evolutionary perspective [32]. Additionally, subliminal stimuli can also have impact on human behavior [33, 34, 35, 36, 37, 38]. In some situations, unconsciously processed stimuli could affect behavioral performance more profoundly than consciously processed stimuli [39]. For instance, a study found that unconsciously processed stimuli could enhance emotional memory after both short and long delays. This result indicated that emotion could enhance memory processing even when the stimuli was encoded unconsciously[36]. Another study found that people's response to painful stimuli was more likely to be affected by unconscious negative emotions than the conscious negative emotions [14, 40]. Therefore, it's imperative to study the processing of subliminal negative stimuli.

In this study, we aimed to compare the neural response to subliminal sad and fearful negative emotions conveyed by eyes on empathy for pain with an ERP technique. We used a typically subliminal affective priming paradigm with an empathy task, which is similar to some previous studies [26, 27, 40], to investigate which negative emotion was highly correlated with empathy for pain automatically. The theory of emotion-sharing indicated that emotion-sharing between the self and others was the base of empathic behavior [41]. When individuals perceived others' emotion, they automatically imitated others' emotions and share the same representation. As we know, the fear emotion means danger and a crisis [42]. It can cause avoidance behaviors. In contrast, sadness can give rise to prosocial behaviors [43]. Furthermore, the priming pictures in one of the previous studies were situational pictures that stimulate participants' own emotions [26]. While our procedure was different in term of that we used different eye emotions as cues to prime empathy for pain. The eye cues were external emotion sources. Hence, we could supply some evidence of finding the suitable emotion from others when empathy for pain occurred to explain the subsequent social behaviors such as helping others or escaping.

As we know, N1 and N2 components reflect early, automatic, affective-sharing processes, people automatically share fearful emotion information rather than sad emotion information when they show empathy for pain [9]. Hence, our hypothesis was that the empathy for pain task with fearful emotion would elicit larger N1 and N2 components than the task with sad

emotion. Some studies showed that the P2 was an index of perceptual processing [44]. Also, humans are sensitive to negative novelty stimuli [45], which captures much more attention and can be reflected by brain activity. Therefore, the negative novelty stimuli would elicit larger amplitudes of ERP components. We therefore hypothesized that the P2 amplitude elicited by sad emotion priming in an empathy task would be larger than that elicited by fear. It meant that people would pay more attention to the empathy task with a sad emotion rather than a fearful emotion. The LPC (late positive component) and P3 component have been shown to index the amount of attentional resources allocated to the stimulus [28], and they indicate top-down processing and elaborate evaluation [16]. The more adequately people evaluate targets, the larger the amplitudes of the P3 and LPC. We hypothesized that the amplitudes of P3 and LPC elicited by painful stimuli were larger than nonpainful stimuli, which meant that processing painful stimuli took more mental resources.

## Materials and methods

### Participants

Nineteen participants with no history of brain injury or mental disorder participated in the experiment. All participants had normal or corrected-to-normal vision, signed the informed consent form and received compensation after the experiment. Four of the participants' data (no obvious classical ERP components for one of them, and valid trials in each condition were 51.8% for the other three) were rejected due to artifacts during electroencephalographic (EEG) recording. Data of 15 participants were retained (6 men, 9 women; 22.33 ± 1.91 years ($M ±$ $SD$)). This study was approved by the Ethics Committee of the Academy of Psychology and Behavior, Tianjin Normal university.

### Experimental design and stimuli

This study used an affective priming paradigm with a 3 priming type (subliminal neutral, sad, fear eye region information) × 2 target type (painful, nonpainful pictures) within-subject design. Therefore, there were 6 conditions in total: Neutral/ Painful, Neutral/ Non-Painful, Sad/ Painful, Sad/ Non-Painful, Fear/ Painful, Fear/ Non-Painful. The priming stimuli used in the experiment were 42 facial black and white pictures (comprising 14 subliminal sad, 14 subliminal fear, and 14 subliminal neutral pictures) selected from the Chinese Affective Face Picture System (CAFPS) [46]. As the total eye region from the CAFPS was not large enough, as well as that some eye region emotions were rated poorly (i.e. some sad eyes did not convey sad emotions sufficiently), a minority of them (7 pictures) were taken from the internet with only remaining the eye region. The gender of the characters in the images for each emotion type was equal. The targets were pictures showing a person's hands in painful or nonpainful situations [47, 48]. The size of eye region pictures was adjusted to 260 pixels in width and 115 pixels in height. The size of masks was 255 × 105 pixels. They were obtained by randomly scrambling 10 × 10-pixel squares on every cropped face with MATLAB2016b. The size of targets was 369 × 287 pixels. The distance between the participants and the computer monitor was approximately 65 cm. The pictures were normalized for size, global contrast and luminance.

Thirty participants assessed the priming pictures prior to the experiment (rated on a 9-point scale). For the level of arousal, both sad ($M ± SD$: 6.71 ± .56) and fearful eye regions (6.84 ± .38) differed significantly in priming from neutral eye regions (3.52 ± .20, $p < .001$), but the difference between the sad and fearful eye regions was not significant ($p = .40$). For level of valence, the sad (2.37 ± .43) and fearful eye regions (2.31 ± .73) differed dramatically from neutral eye regions (4.30 ± .56, $p < .001$), but the difference between the sad and fearful eye regions was not significant ($p = .79$). Another 29 participants assessed the target pictures

(rated on a 4-point scale). The pain intensity of the painful pictures ($M \pm SD$: 2.16 ± .22) and the nonpainful pictures (.30 ± .19) were significantly different ($p < .001$).

## Experimental procedure

The experimental procedure in each trial was as follows. A fixation was presented in the center of the screen for 500 ms, followed by a random blank screen for 400~600 ms. The priming pictures were then presented for 16 ms. Subsequently, the scrambled mask pictures were presented for 184 ms follow by a blank screen of 200 ms [40]. After which, the target pictures were displayed for 1500 ms until a response was given. Participants were told to judge whether the targets were in pain or not. There was a blank interval presented for 1000 ms between each trial (see Fig 1). Simultaneously, the brain activity was recorded using Curry8 (Neuroscan, USA). The priming reached a subliminal effect through a series of operations consisting of short-term presentation of the priming pictures with masks subsequently. After the EEG session, participants were asked whether they saw the priming pictures or their type clearly before the mask. They all reported that they were not aware of the emotion type. Therefore, the priming pictures were at an unconscious level according to their subjective reports [40]. Consequently, this operation ensured that the priming was subliminal.

The procedure was divided into 4 blocks. Each block consisted of 126 trials, hence 84 trials in each condition. Each type of eye region pictures repeated 12 times, and each of the target pictures repeated 6 times. The stimuli in every block were pseudorandomly presented. The response buttons were counterbalanced across participants, and the button press model was

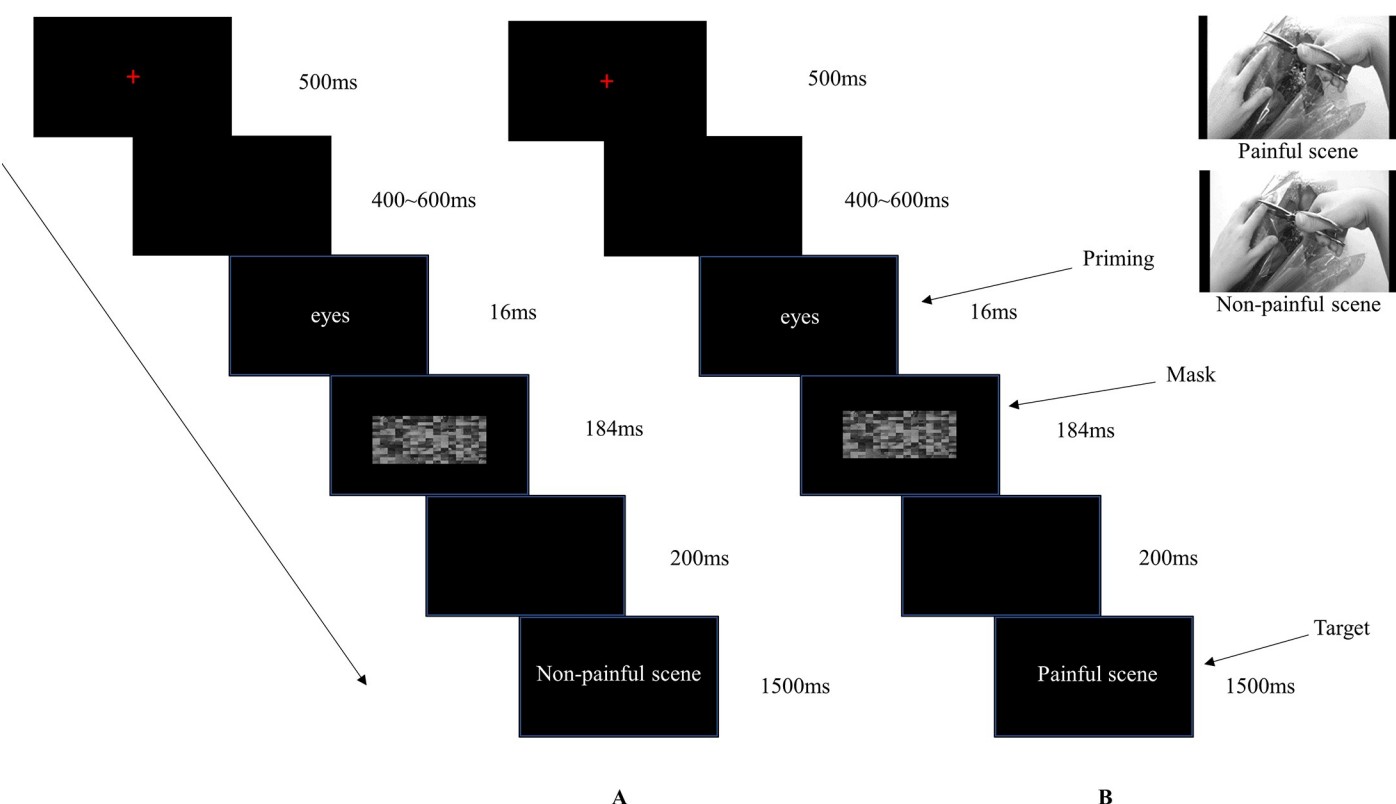

**Fig 1. The procedure of the effect of subliminal emotional information from eyes on empathy for pain.** Panel A showed the target illustrated a non-painful scene. Panel B showed the target illustrated a painful scene. Participants were asked to judge whether the target was painful or not.

also counterbalanced across different participants using F for painful targets and J for non-painful targets or J for painful targets and F for nonpainful targets.

After the EEG session, all participants were asked to complete the Chinese version of Inter-personal Reactivity Index scale (IRI-C scale) [3, 49], which contained 22 items. A 5-point Likert scale was used for responses, ranging from inappropriate (0 point) to very appropriate (4 points). 5 items were scored in reverse. In this questionnaire, empathy was divided into 4 parts: perspective-taking (tendency to spontaneously adopt the psychological point of view of others), empathy concern ("other-oriented" feelings of sympathy and concern for unfortunate others), fantasy (tendency to transpose themselves imaginatively into the feelings and actions of fictitious characters in books, movies, and plays) and personal distress ("self-oriented" feel-ings of personal anxiety and unease in tense interpersonal settings). Correlations between the personal distress subscale and ERP components was conducted in order to find out the rela-tionship between state empathy in task and trait empathy [30].

## Devices and recording

The EEG was continuously recorded from 64 scalp electrodes mounted on an elastic cap in accordance to the extended 10–20 system, with the addition of two mastoid electrodes. The electrode at the left mastoid was used as the recording reference and transformed to the aver-age of the two mastoids offline. The electrode impedance was kept at less than 5 kΩ. Eye blinks and vertical eye movements were monitored with electrodes located above and below the left eye. The horizontal electro-oculogram was recorded from electrodes placed 1.5 cm lateral to the left and right external canthi. The EEG was amplified (bandpass: 0.05–400 Hz) and digi-tized at a sampling rate of 1000 Hz.

## Data analysis

EEG data were preprocessed and analyzed using the functions in Curry8 software. To achieve more valid trials, some frontal electrodes (FP1 FPz FP2 AF3 AF4) were excluded for drifting and some electrodes over the ears (F7 FT7 T7 TP7 F8 FT8 T8 TP8) were excluded due to low signal quality. Additionally, the ineffective channel CP5 was deleted. The EEG data were re-referenced to linked offline mastoids. A digital filter at a low-pass of 30 Hz was then applied. Trials exceeding 100 μv at any electrode were excluded from the average. Time windows of 600 ms before and 1000 ms after the onset of the painful and nonpainful target pictures were segmented from EEG data. The time of target onset was the zero point. All epochs were base-line-corrected with respect to the mean voltage over the 600 ms to 400 ms preceding the onset of painful and nonpainful targets. Baseline correction was performed prior to fixation to avoid distortion of the baseline by visually evoked potentials to the mask [50, 51]. An additional ANOVA in the time interval between -400 ms and -200 ms was performed to exclude any physical difference in the stimuli that may have compromised possible priming effects [50, 51]. This analysis did not reveal any significant effects ($ps > .09$).

According to the grand-averaged ERP pictures, the topographical distribution and the liter-ature [9, 30, 52, 53], five regions of interest from the frontal (the mean amplitudes of F3, Fz and F4), frontal-central (the mean amplitudes of FC3, FCz and FC4), central (the mean ampli-tudes of C3, Cz and C4), central-parietal (the mean amplitudes of CP3, CPz and CP4) and parietal (the mean amplitudes of P3, Pz and P4) regions were chosen. Analyses were conducted using the peak amplitudes of N1 (100~180 ms), P2 (180~240 ms), N2 (240~300 ms) and P3 (300~430 ms) and the mean amplitude of the LPC (430~650 ms) after the onset of the painful or nonpainful pictures [9, 16, 28]. A three-way, repeated-measures ANOVA with priming type (subliminal neutral, sad, fear eye region information), target type (painful, nonpainful

pictures) and regions of interest was performed. Specifically, five electrodes from the midline (Fz, FCz, Cz, CPz and Pz) were chosen for LPC [40]. Degrees of freedom for the *F*-ratio were corrected using the Greenhouse-Geisser method. Statistical differences were corrected according LSD post hoc correction.

Moreover, we conducted the correlations between PD ratings and pain effect (subtracting ERP elicited in the nonpainful condition from ERP elicited in the painful condition) to investigate whether the pain effect was correlated with the PD subscale of the IRI-C [30].

## Results

### Behavioral results

A three-way repeated measures ANOVA was conducted to examine the reaction time (RT) differences between the experimental conditions. The results are depicted in Fig 2.

The main effect of priming type was significant ($F(2,28) = 28.21$, $p < .001$, $\eta^2_p = .67$). The RT for the neutral eye emotion ($M = 800$ ms, $SE = 20$) was faster than the fearful eye emotion ($M = 816$ ms, $SE = 22$) ($p < .01$) and the sad eye emotion ($M = 827$ ms, $SE = 21$) ($p < .001$), and the RT for the fearful eye emotion was faster than the sad eye emotion ($p < .01$). The main effect of target type was significant ($F(1,14) = 7.20$, $p = .02$, $\eta^2_p = .34$). The RT for the painful pictures ($M = 797$ ms, $SE = 19$) was faster than for nonpainful pictures ($M = 832$ ms, $SE = 24$). There was an interaction between priming type and target type ($F(2,28) = 7.99$, $p < .01$, $\eta^2_p = .36$), suggesting that the RT for the neutral eye emotion ($M = 776$ ms, $SE = 18$) was faster than the fearful eye emotion ($M = 795$ ms, $SE = 20$) ($p < .01$) in the painful situation, and there was also a significant difference between the neutral and sad eye emotions ($M = 820$ ms, $SE = 20$) ($p < .001$) and the fearful and sad eye emotions ($p = .001$) in the painful situation. Other interactions were not significant ($ps > .07$).

### ERP results

**N1.** Only one significant main effect of region was observed ($F(4,56) = 48.06$, $p < .001$, $\eta^2_p = .77$). N1 was mainly distributed in the frontal and frontal-central regions ($M \pm SE$: -6.24 $\pm$ .71μV, -5.76 $\pm$ .65μV, -3.60 $\pm$ .38μV, -.31 $\pm$ .21μV and 2.08 $\pm$ .53μV). The main effect of

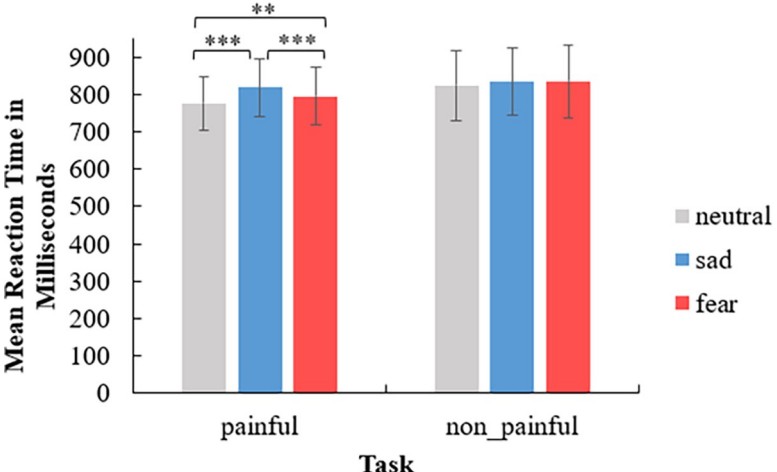

**Fig 2. Behavioral results under six experimental conditions.** The error bars representing standard errors of the mean. The symbol of "**" means $p < .01$. "***" means $p \leq .001$.

priming type was insignificant ($F(2,28) = 1.12$, $p = .34$). The main effect of target type was not significant ($F(1,14) = .36$, $p = .56$). All the interactions were not significant ($ps > .16$).

**P2.** The main effect of region was significant ($F(4,56) = 33.02$, $p < .001$, $\eta^2_p = .70$). P2 was mainly distributed in the parietal regions ($M \pm SE$: -2.16 ± .63µV, -2.00 ± .56µV, -.78 ± .31µV, 1.56 ± .20µV and 4.70 ± .59µV). The main effect of priming type was insignificant ($F(2,28) = .54$, $p = .59$). The main effect of target type was insignificant ($F(1,14) = .22$, $p = .65$). A significant interaction of priming type × target type ($F(2,28) = 3.85$, $p = .03$, $\eta^2_p = .22$) was observed, suggesting that the sad eye emotion($M \pm SE$: .33 ± .18µV) elicited a larger P2 amplitude than the fear (.15 ± .21µV) under the painful situation($p = .02$), but there was no significant difference between the neutral(.28 ± .21µV) and the fear eye emotion ($p = .08$) as well as the sad and the neutral eye emotion ($p = .52$). Other interactions were not significant (ps > .61) (see Fig 3 and Fig 4).

**N2.** There was only one significant main effect of region ($F(4,56) = 60.86$, $p < .001$, $\eta^2_p = .81$), suggesting that N2 was mainly distributed in the frontal and frontal-central regions ($M \pm SE$: -6.59 ± .67µV, -6.04 ± .57µV, -3.49 ± .35µV, .07 ± .30µV and 3.12 ± .59µV). The main effect of priming type was insignificant ($F(2,28) = .54$, $p = .59$). The main effect of target type was also insignificant ($F(1,14) = .08$, $p = .78$). All the interactions were not significant ($ps > .47$)

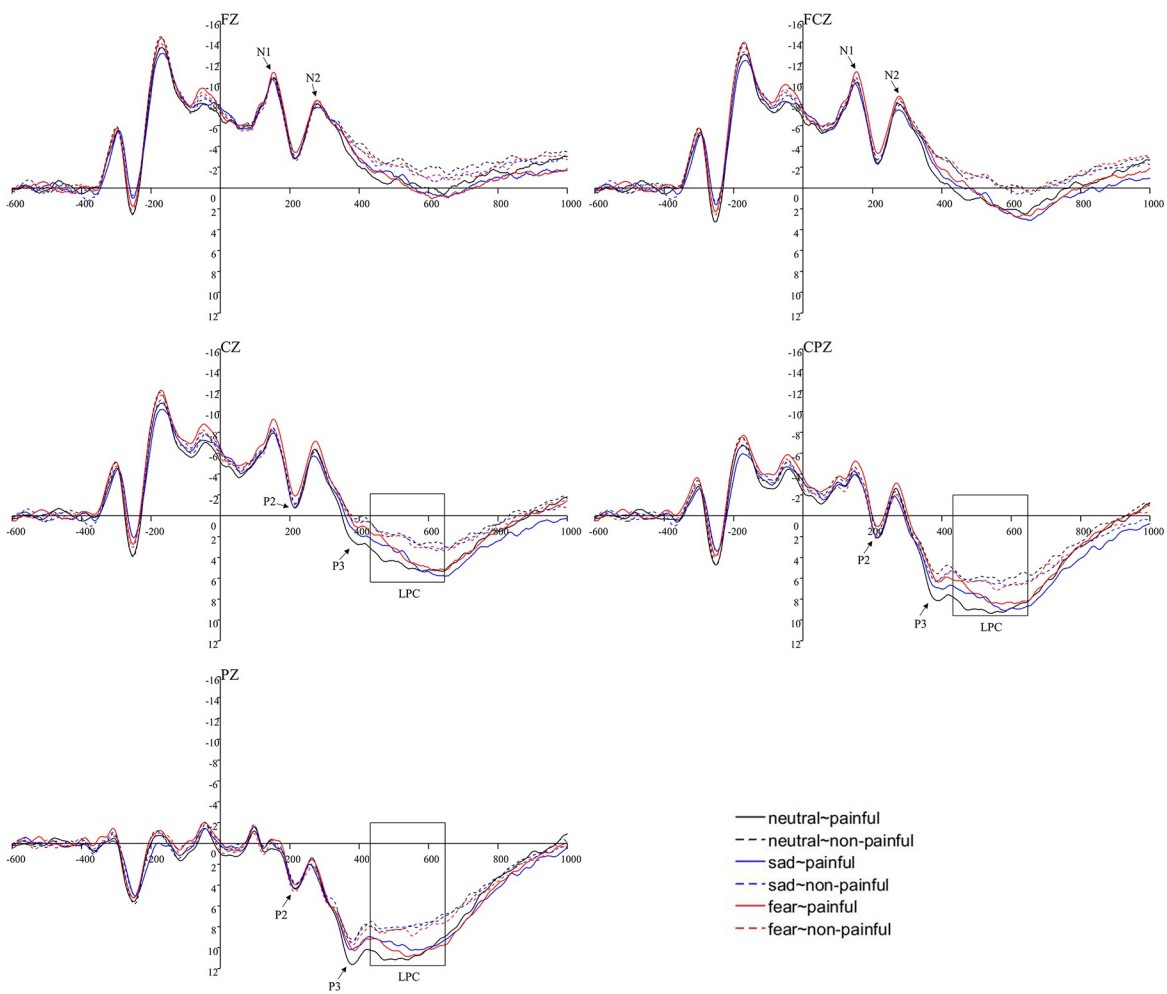

**Fig 3. Illustration of ERP results at representative electrodes.**

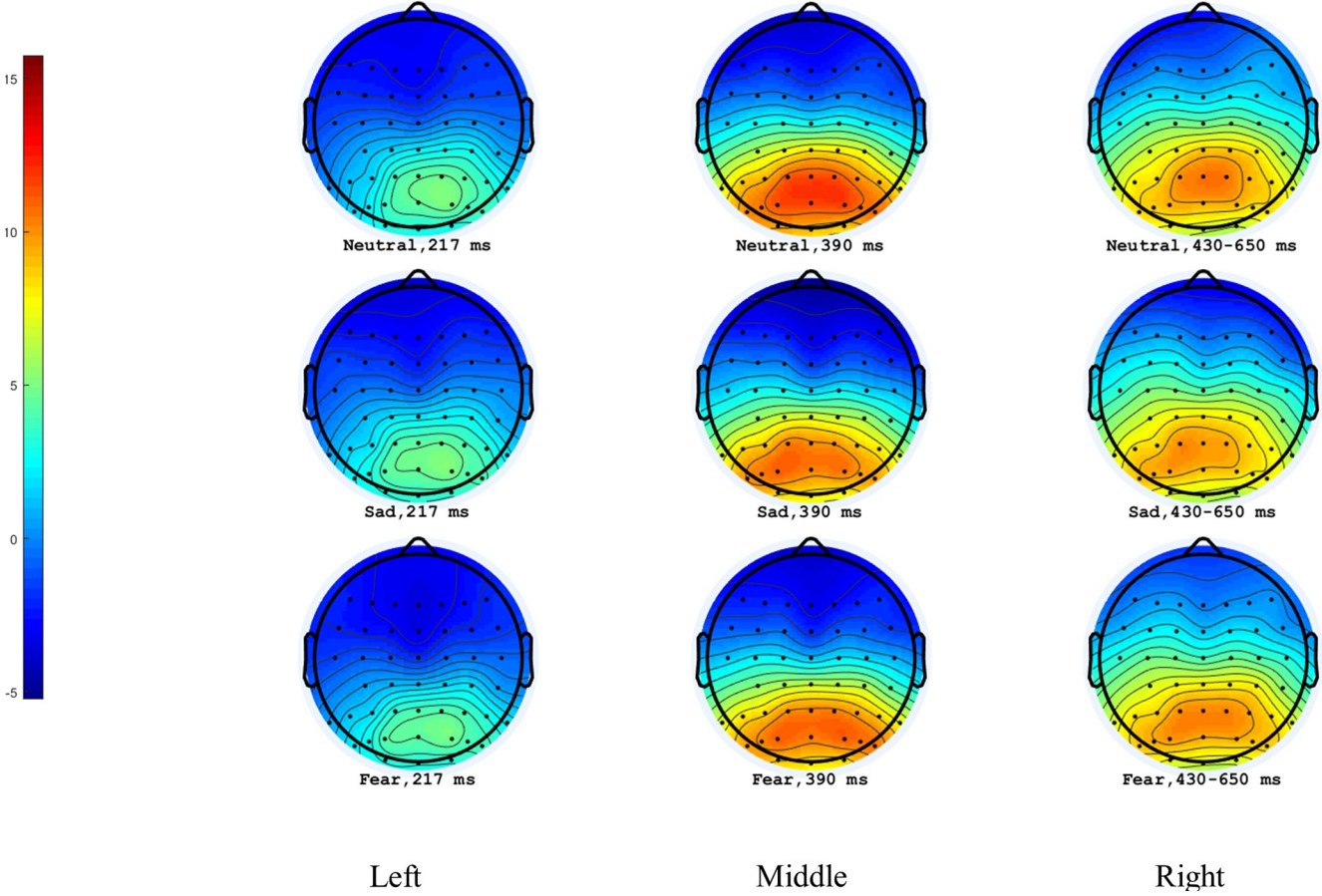

|      | Left | Middle | Right |
|------|------|--------|-------|

**Fig 4. Topographical maps of P2, P3 and LPC for the painful conditions.** Left panel represented P2 peak amplitudes. Middle panel represented P3 peak amplitudes. Right panel represented LPC mean amplitudes.

**P3.** The significant main effect of region was observed ($F(4,56) = 73.86$, $p < .001$, $\eta^2_p = .84$), suggesting that P3 was mainly distributed in the parietal regions ($M \pm SE$: -4.68 ± .76μV, -3.82 ± .64μV, -1.32 ± .34μV, 2.77 ± .31μV and 6.80 ± .51μV). The main effect of priming type was insignificant ($F(2,28) = .77$, $p = .47$). The main effect of target type was significant ($F(1,14) = 7.46$, $p = .02$, $\eta^2_p = .35$), suggesting that comparing to nonpainful stimuli($M \pm SE$: -.11 ± .21μV), a larger amplitude was elicited by painful pictures (.01 ± .20μV). All the interactions were not significant ($ps > .29$) (Figs 3 and 4).

**LPC.** The main effect of the electrode was significant ($F(4,56) = 52.95$, $p < .001$, $\eta^2_p = .79$), suggesting that LPC was largest at $P_Z$ ($M = 4.35$ μV, $SE = .54$). The main effect of priming type was not significant ($F(2,28) = 1.18$, $p = .32$). The main effect of target type was significant ($F(1,14) = 10.52$, $p = .01$, $\eta^2_p = .43$) with painful pictures eliciting a larger amplitude ($M \pm SE$: -.63 ± .34 μV) than nonpainful pictures (-.98 ± .28 μV). No interactions were significant ($ps > .13$) (Figs 3 and 4).

### Correlations between the IRI-C and ERP

To investigate whether the pain effect (subtracting ERP elicited in the nonpainful condition from ERP elicited in the painful condition) was correlated with the PD subscales of the IRI, we calculated the correlations between PD ratings ($M \pm SD$: 1.40 ± .77) and the pain effect from

the frontal, frontal-central and the central electrodes (i.e., F3, Fz, F4, FC3, FCz, FC4, C3, Cz and C4) based on the distributions of N1 and N2. These analyses revealed significant correlations in the N1 time window (Cz: $r(15) = .51$, $p = .05$ for PD) and N2 time window (F3: $r(15) = .61$, $p = .02$ for PD).

## Discussion

Previous studies showed the significance of subliminal stimuli. For instance, subliminal stimuli that failed to reach awareness may affect performance more effectively than stimuli that do reach awareness [54]. Invisible stimuli that are irrelevant to task are more disturbing than visible distractors because of failed communication in regions of the brain that normally suppress such responses [54]. Meanwhile, a study found that the affectively significant visual stimuli had privileged access to consciousness [55]. Our research initially used subliminal eye region information as the experimental stimulus to explore its effect on empathy for pain. The eye region was found to be an important area for conveying facial expression [25]. The subliminal paradigm in the present study was used to highlight the automatic processing of eye cues. Furthermore, unlike previous studies that roughly divided the experimental materials into positive, neutral, and negative emotion types, the current research divided the types of negative emotions more elaborate. We chose neutral, sad and fearful emotions because sad and fearful emotions have great significance for human survival. Sadness can give rise to prosocial behaviors [43] and fear can cause avoidance behaviors. Therefore, in this study, we compared the neural response to subliminal sad and fearful negative emotions conveyed by eye cues on empathy for pain by measuring the ERP responses.

The behavioral results showed that people reacted to the painful scene more slowly with the subliminal sad eye emotion than with the other two emotions. This finding indicated that people were more affected by sad eye emotion and sad eye emotion information required more attention. It seemed that people could not extract their attention from the sad emotion immediately during the empathy task. The sad emotion interfered with their responses to pain judgements [56]. Consequently, participants reacted more slowly to painful scenes while viewing the subliminal sad eye region information. On the other hand, explanation of negative priming effect supposed that during the process of subliminal priming, when the primes and the targets were related, automatic spreading activation from primes did happen and it might interfere with the process and comparison of the subsequent targets owing to the short interval in subliminal paradigms. When the primes and the targets were unrelated, they rarely matched and interfered with each other, so that the response was quicker [57].

The ERP results demonstrated that the P2 amplitude was less positive for the fear condition relative to the sadness condition for painful scenes. Neither sad nor fearful conditions were significantly different from neutral conditions for painful scenes. First of all, the P2 component is an index of extraction of emotional salience from stimuli [58]. It seemed that sad emotion from the eye regions stood out from the other two emotions during the empathy for pain task. Further, the P2 is also an index of attention [59]. Attentional bias happens automatically. Comparing to subliminal fearful eye regions, the empathy task was modulated and attracted more by subliminal sad eye regions. This meant people focused more on the empathy task primed by subliminal sad eye regions than that primed by fearful eye regions. Furthermore, sad eye emotions induce human's prosocial behaviors to some extent [43]. However, fear emotion often implies a crisis [42]. Hence, there will be more costs involved in human's exploration [60]. As a result, more attentional resources were diverted to aid in processing the sad eye emotion, and a larger P2 amplitude during empathy for pain was evoked.

Moreover, the task setting might cause participants some psychological stress and emotional burden [61]. Participants were asked to make a pain judgement within a certain period of time. They might imagine that they were experiencing the situation of the protagonist in a social scene. In this way, participants might have experienced negative impression and evaluation of the neutral eye emotions, which initially represented calm emotions. Consequently, the neutral emotion, to some extent, was more threatening and dominant rather than purely neutral [62]. Furthermore, the degree of threat was a little weaker than the fearful eye emotion. There was no significant difference between them accordingly. There was also no significant difference between the prosocial mental state [43, 63] elicited by the sad eye emotion and the dominant mental state induced by the neutral eye emotion. Maybe, the brain activities of the two emotions on empathy for pain were at a similar level under our task setting.

Additionally, this study found the amplitude elicited by painful pictures was larger than nonpainful pictures for late-controlled P3 and LPC. These two components were related to cognitive evaluation and involved elaborate processing based on previous results [9, 16, 26, 64]. The behavioral results demonstrated that people reacted more quickly than with the non-painful situation. This finding may be related to the allocation of human attentional resources [26]. Human attentional resources are limited, and they have attentional bias. The painful scenes were more novel and important for human survival. Also, due to the task setting, participants were asked to judge whether the stimuli were in pain or not. Painful scenes were much more varied than nonpainful scenes. They therefore captured more attentional resources [65], and received more cognitive resources [26]. It might be the reason why the amplitude elicited by the painful stimuli was significantly larger than that elicited by the nonpainful stimuli.

Lastly, we explored the relationship between IRI-C and ERP components. According to a previous study, PD ratings were used to reflect automatic affective sharing of others' emotional experience [30, 66, 67, 68]. The N1 and N2 have been suggested to be markers of early automatic emotional sharing [9, 30]. Therefore, we calculated the correlations between them. We found that the different waves (subtracting ERP elicited in the nonpainful condition from ERP elicited in the painful condition) of N1 and N2 were significantly correlated with PD sub-scale ratings. The larger the difference between painful and nonpainful conditions, the higher the PD score elicited. This significant correlation suggested that N1 and N2 reflected the level of discomfort induced by affective sharing of another's pain [30]. The correlation between the PD rating and the early ERP reaction to the pain effect suggested that the empathic behavioral response had a neural basis.

## Limitations

There are some possible directions for further study, such as the influence of different arousal levels from the same emotional priming on empathy for pain. Additionally, the source of the priming type (eye region emotion from different interpersonal distances) could be another influential factor in empathy for pain. There is still more space for exploration.

## Conclusions

Together, these findings demonstrated that subliminal eye emotions significantly affected empathy for pain. Compared with the subliminal fearful eye stimulus, the subliminal sad eye stimulus had a greater impact on empathy for pain, which supported the theory of emotional sharing. The perceptual level of pain was deeper in the late controlled processing stage.

## Acknowledgments

The authors would like to acknowledge all the participants in this study and Tianjin Philosophy and Social Science Projects in China. (TJJX17-010).

## Author Contributions

**Conceptualization:** Juan Song.

**Methodology:** Yanqiu Wei.

**Supervision:** Juan Song.

**Validation:** Juan Song.

**Writing – original draft:** Yanqiu Wei.

**Writing – review & editing:** Juan Song, Han Ke.

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
