## [Decision Letter · Decision Letter 0]

14 Aug 2019

PONE-D-19-16381

The effect of emotional information from eyes on empathy for pain A subliminal ERP Study

PLOS ONE

Dear Dr. Song,

Thank you for submitting your manuscript to PLOS ONE. After careful consideration, we feel that it has merit but does not fully meet PLOS ONE’s publication criteria as it currently stands. Therefore, we invite you to submit a revised version of the manuscript that addresses the points raised during the review process.

We would appreciate receiving your revised manuscript by Sep 24 2019 11:59PM. To enhance the reproducibility of your results, we recommend that if applicable you deposit your laboratory protocols in protocols.io, where a protocol can be assigned its own identifier (DOI) such that it can be cited independently in the future. For instructions see: http://journals.plos.org/plosone/s/submission-guidelines#loc-laboratory-protocols

We look forward to receiving your revised manuscript.

Kind regards,

Kai Wang

Academic Editor

PLOS ONE

Journal Requirements:

1. We suggest you thoroughly copyedit your manuscript for language usage, spelling, and grammar. If you do not know anyone who can help you do this, you may wish to consider employing a professional scientific editing service.  

2. Please include a separate caption for each figure in your manuscript.

3. Please upload a copy of Supporting Information Figures S1-S4 which you refer to in your text on page 10

4. We note that Figure(s) [1] in your submission contain copyrighted images. All PLOS content is published under the Creative Commons Attribution License (CC BY 4.0), which means that the manuscript, images, and Supporting Information files will be freely available online, and any third party is permitted to access, download, copy, distribute, and use these materials in any way, even commercially, with proper attribution. For more information, see our copyright guidelines: http://journals.plos.org/plosone/s/licenses-and-copyright.

1.    You may seek permission from the original copyright holder of Figure(s) [1] to publish the content specifically under the CC BY 4.0 license.

Reviewers' comments:

Reviewer's Responses to Questions

**Comments to the Author**

1. Is the manuscript technically sound, and do the data support the conclusions?

Reviewer #1: Yes

Reviewer #2: Yes

2. Has the statistical analysis been performed appropriately and rigorously? 

Reviewer #1: No

Reviewer #2: No

3. Have the authors made all data underlying the findings in their manuscript fully available?

Reviewer #1: Yes

Reviewer #2: Yes

4. Is the manuscript presented in an intelligible fashion and written in standard English?

Reviewer #1: No

Reviewer #2: Yes

5. Review Comments to the Author

Reviewer #1: The research article by Song and co-authors addresses the very interesting question if there is an association between eye region information and empathy for pain. This study demonstrated that subliminally conveyed eye emotion affected the viewer’s empathy for pain. Compared with the subliminal fearful eye stimulus, the subliminal sad eye stimulus had a greater impact on empathy for pain. The perceptual level of pain was deeper in the late controlled processing stage. Nevertheless I have some important points that I believe should be changed to make the manuscript clearer.

Specific comments:

ABSTRACT

a. In the body text the authors report the relationship between IRI-C and ERP components. However, this is not expressed in the abstract. This point should be added to make this paper more preciseness in the abstract.

b. Some abbreviations in the abstract should be clearly indicated.

Introduction

a. Some abbreviations are not given, such as “temporal poles” and “ inferior frontal gyrus”.

b. What is the meaning of “N170”? I noticed that this is explained in your 16th reference. Maybe you can explain what it means in your article.

c. In the third sentence of the sixth paragraph in your introduction, the conjunction “Nevertheless” may not be appropriate.

d. In the sixth paragraph in your introduction, we should start by writing about what others have investigated, and then write something about what research is fully investigated and what we can do in the present study.

e. The two sentences “Then, how do humans perceive others’ pain in completely different situations? It is still unknown and needs to be further explored.” may be better expressed in a declarative sentence.

Methods

a. The authors write "Nineteen participants with no history of brain injury or mental disorder participated in the experiment." Then, we can see “Thirty participants” and “another 29 participants”coming up in the Experimental Design and Stimuli part. Did the authors make some writing mistakes or some other reasons?

b. What is “IRI-C scale”? I did not find the full name of this abbreviation, which should be indicated in the text. There exists other improper abbreviation, for example the “PD”.

c. Why only make a correlation analysis between the pain effect and the Personal Distress (PD) subscales of the IRI-C? How about other scales of the IRI-C?

Results

- The abbreviation “RT” is not clear.

Discussion

a. The authors write “Fear indicates danger and a crisis, which can cause avoidance behaviors.”“ This finding might be caused by people being more affected by sad eye emotion and sad eye emotion information requiring more attention.” Is there any grammatical error here?

b. The authors write “How do humans automatically perceive others’ pain in these different situations? We still need to explore this question”. This point may be not appreciated. I think it can be replaced by your goal and hypothesis of your research.

c. “Additionally, P2 is an index of attention [55]. It is enhanced with greater attention.” These two sentences have the same meaning, change the expression of a subordinate clause or delete one sentence.

d. Psychology is very gender-biased. Whether gender influences the results in this study is worth further discussion. All participants have no history of brain injury or mental disorder in the study, does this phenomenon still exist in other groups or in other age groups?

Reviewer #2: This paper explored the effect of subliminal sad and fearful information conveyed by eyes on pain empathy by ERPs, and found that sad lead to slower response and greater P2 amplitude. The focus of the paper are interesting, the methods and used and the logic are reasonable, the structure of the article is clear. The following problems need further addressed.

Major points:

1. This article should focus on the modulation of subliminal emotional primes on pain empathy, so authors need to pay more attention to the interactions between the primes and targets. For example, authors only report P3 and LPC are affected by painful / nonpainful targets, can subliminal emotion be modified this effect? The results and discussion are less concerned about interactions, the reported interactions in the results are also poorly explained.

2. The author used the subliminal stimuli as primes, but in the introduction part, they did not present enough reasons about the significance of subliminal emotion and the difference between subliminal and supraliminal stimuli. The discussion part also lack the related explanation.

3. Methods and Results: the number of subjects (only 15 subjects’ results) was relatively small. In addition to arousal and valence, did the emotional types (sad, fearful or neutral?) be evaluated by subjects? Authors report interactions in the results, but do not report clearly how they interact in subsequent analyses from the view of two independent variables, so that readers can understand.

4. In view of the current results, there is a suggestion for data analysis. When analyzing ERP data, the author conducted repeated measurement analysis of brain region×primes (sad, fearful and neutral)× target type (painful or nonpainful), and the main results focused on the main effect of the region, while other results were not good enough. The ERP components differed in different brain regions, which is natural an easily understood results. Therefore, the analysis of specific ERP component can be conducted by analyzing the differences in brain regions first, and then analyze the electrodes of the most obvious region with prime types× target types measurements, which can more effectively reflect the author's intention.

Minor points:

Figure 1: It is suggested to replace the eyes, mask and target of the actual material samples

Figure 2: To add SE as error bar to the bar of reaction time. The unit and description should be added on the axis. The vertical axis should start from 0, and statistics significance is suggested to be added.

Figure 3：The ERP figure is not clear enough to see the detail including time axis. There are six lines in the ERP figure. It is recommended to take the peaks and draw a bar chart to express the results more clearly.

6. PLOS authors have the option to publish the peer review history of their article (what does this mean?). If published, this will include your full peer review and any attached files.

Reviewer #1: No

Reviewer #2: Yes: Bin Xuan

---

## [Author Response · Author response to Decision Letter 0]

23 Oct 2019

Response to Editor and Reviewers(Please see the file "Response to Editor and Reviewers"in the final PDF. The colored part of revision can be seen in the file.)

Response to Editor and Reviewers

Dear editor and reviewers,

Thank you very much for inviting us to revise and resubmit our manuscript “The effect of emotional information from eyes on empathy for pain: A subliminal ERP Study”. We appreciate this opportunity. The comments raised by the reviewers were very thoughtful and constructive. Based on these comments, we have revised our manuscript and we believe that this revision is a substantially stronger contribution accordingly. We have prepared this cover letter to summarize how we have addressed these comments in our revision. Please see below for detailed responses. The issues raised in your editorial letter is reproduced in italics, followed by our response in each case.

We look forward to your reply. We have already sent our manuscript to AJE for language editing before our submitting. Also, for this revision, we accepted suggestions from one native speaker of English and one professional visitor study abroad about our language again. However, if needs, please give us another chance to edit it. See the revision of language in words highlighted in grey and some red words.

We are confident about our results and we also tried the methods the review suggested. If there is any thing you think we should revise or do more, please give us another chance to do it. We think our work is quiet creative and new in empathy study field. So it has its own value to be published. And it will really be our great pleasure if we can have it published in your journal with the help and suggestions of our editor and reviewers. 

Thank you so much for your work and patience. Every details we revised were highlighted in the revised version of manuscript. 

And we also revised the figures according to the advice.

If there is anything we did not revise or understand, please contact us and give us another chance, We will really appreciate.

Sincerely,

Juan Song on behalf of all authors

Reviewer #1: The research article by Song and co-authors addresses the very interesting question if there is an association between eye region information and empathy for pain. This study demonstrated that subliminally conveyed eye emotion affected the viewer’s empathy for pain. Compared with the subliminal fearful eye stimulus, the subliminal sad eye stimulus had a greater impact on empathy for pain. The perceptual level of pain was deeper in the late controlled processing stage. Nevertheless, I have some important points that I believe should be changed to make the manuscript clearer. 

Specific comments: 

Thank you so much for your suggestions.Please see the revised version of manuscript and manuscript with track changes. We also have Figure1,2,3,4(revised) version.

ABSTRACT 

a. In the body text the authors report the relationship between IRI-C and ERP components. However, this is not expressed in the abstract. This point should be added to make this paper more preciseness in the abstract.

We really thank the Reviewer’s advice on the preciseness of the abstract. We have revised it: Moreover, the subjective ratings of Personal Distress (PD) (one of the dimensions in Chinese version of Interpersonal Reactivity Index scale) predicted the pain effect in empathic neural responses in the N1 and N2 time window.

“Abstract

Facial expressions are deeply tied to empathy, which plays an important role during social communication. The eye region is effective at conveying facial expressions, especially fear and sadness emotions. Further, it was proved that subliminal stimuli could impact human behavior. This research aimed to explore the effect of subliminal sad, fearful and neutral emotions conveyed by the eye region on a viewer’s empathy for pain using event-related potentials (ERP). The experiment used an emotional priming paradigm of 3 (prime: subliminal neutral, sad, fear eye region information) × 2 (target: painful, nonpainful pictures) within-subject design. Participants were told to judge whether the targets were in pain or not. Results showed that the subliminal sad eye stimulus elicited a larger P2 amplitude than the subliminal fearful eye stimulus when assessing pain. For P3 and late positive component (LPC), the amplitude elicited by the painful pictures was larger than the amplitude elicited by the nonpainful pictures. The behavioral results demonstrated that people reacted to targets depicting pain more slowly after the sad emotion priming. Moreover, the subjective ratings of Personal Distress (PD) (one of the dimensions in Chinese version of Interpersonal Reactivity Index scale) predicted the pain effect in empathic neural responses in the N1 and N2 time window. The current study showed that subliminal eye emotion affected the viewer’s empathy for pain. Compared with the subliminal fearful eye stimulus, the subliminal sad eye stimulus had a greater impact on empathy for pain. The perceptual level of pain was deeper in the late controlled processing stage.”

b. Some abbreviations in the abstract should be clearly indicated.

Thanks for your nice advice. The abbreviations (ERP, LPC) in our abstract has been clearly indicated in our manuscript.

“Abstract

Facial expressions are deeply tied to empathy, which plays an important role during social communication. The eye region is effective at conveying facial expressions, especially fear and sadness emotions. Further, it was proved that subliminal stimuli could impact human behavior. This research aimed to explore the effect of subliminal sad, fearful and neutral emotions conveyed by the eye region on a viewer’s empathy for pain using event-related potentials (ERP). The experiment used an emotional priming paradigm of 3 (prime: subliminal neutral, sad, fear eye region information) × 2 (target: painful, nonpainful pictures) within-subject design. Participants were told to judge whether the targets were in pain or not. Results showed that the subliminal sad eye stimulus elicited a larger P2 amplitude than the subliminal fearful eye stimulus when assessing pain. For P3 and late positive component (LPC), the amplitude elicited by the painful pictures was larger than the amplitude elicited by the nonpainful pictures. The behavioral results demonstrated that people reacted to targets depicting pain more slowly after the sad emotion priming. Moreover, the subjective ratings of Personal Distress (PD) (one of the dimensions in Chinese version of Interpersonal Reactivity Index scale) predicted the pain effect in empathic neural responses in the N1 and N2 time window. The current study showed that subliminal eye emotion affected the viewer’s empathy for pain. Compared with the subliminal fearful eye stimulus, the subliminal sad eye stimulus had a greater impact on empathy for pain. The perceptual level of pain was deeper in the late controlled processing stage.”

Introduction 

a. Some abbreviations are not given, such as “temporal poles” and “inferior frontal gyrus”.

Really thanks a lot for your clear advice. We revised it clearly in our new manuscript.

Please see in the second paragraph of Introduction:

“Empathy for pain has been the major focus of research devoted to empathy in social neuroscience and other related fields, making it the most dominant neuroscientific domain in the study of empathy [7]. A review paper summarized research in this field for the past twenty years, it was found that the theme of empathy research gradually shifted from early personality traits, attitudes, and emotions to social cognition [8]. With the development of technology, research on the cognitive neuroscience of empathy became a hot topic. On one hand, some researchers thought that the medial prefrontal cortex (mPFC), superior temporal sulcus (STS), temporal poles (TP), and ventromedial cortex (VM cortex) were involved in cognitive empathy, and the amygdala, insula and inferior frontal gyrus (IFG) were involved in affective empathy. On the other hand, the right temporoparietal junction (rTPJ), the insula and the anterior midcingulate cortex (aMCC) were thought to be related to empathy for pain [9]. The aMCC and the insula were the “shared neural circuit” of empathy for pain [10]. In other research, the prefrontal cortex (PFC) and the anterior cingulate cortex (ACC) were closely related to empathy for pain [11]. Some new research found that the fusiform face area (FFA) was involved in empathy tasks [12]. The mirror neuron, “neural bridge” in social communication, occupies an important place in empathy [13].”

b. What is the meaning of “N170”? I noticed that this is explained in your 16th reference. Maybe you can explain what it means in your article.

That’s a creative question. But we did not explore N170 ERP component in this research. So we did not describe it more. But we also add some of its function as you suggested. As we know, N170 is sensitive to eye information. For example, isolated eye regions elicit N170 amplitudes and delayed latencies compared to faces (Parkington & Itier, 2018). Maybe, our future research can mainly explore whether the eye region information as a priming cue could elicit the N170.

Please see the third paragraph of Introduction.

“As we know, the face can provide important social information and reflect potential changes in the environment [14, 15]. People can infer and understand others’ inner state immediately by observing facial expressions. A study found that individuals with high empathy paid attention to various negative facial expressions (angry and afraid faces) more than those with low empathy from very-early stage (reflected in the N170) to late-stage (reflected in the LPP) processing of faces [16]. N170 is a negative component elicited by face’s feature and is sensitive to eye information. For example, isolated eye regions elicit N170 amplitudes and delayed latencies compared to faces. Hence, there is a close relationship between empathy and facial expression.” 

c. In the third sentence of the sixth paragraph in your introduction, the conjunction “Nevertheless” may not be appropriate.

We appreciate that the reviewer’s spirit of preciseness. We have replaced it with “Further”.

“In summary, the eye region was found to be the important area for conveying facial expression [25]. In previous research, the ERP research concerning empathy for pain mostly used pictures and videos describing hands or feet in painful situations, or they used a painful facial expression to explore the subjects’ index of brain activity and behavioral changes when observing others’ pain [26, 27, 28, 29, 30]. To date, there have been a few ERP studies comparing the effect of negative emotions on empathy for pain [26, 27, 31]. For example, one supraliminal ERP study looking at the effect of eye emotional information on empathy for pain found a significant difference on P3 component induced by different eye conditions. The amplitude in neutral eyes condition was greater than that in sad eyes condition and fear eyes condition. Different supraliminal emotional information expressed by eye region affects late processing of empathy for pain [31]. Further, negative emotions have a profound influence on human survival and are important for social adaption from an evolutionary perspective [32]. Additionally, subliminal stimuli can also have impact on human behavior [33, 34, 35, 36, 37, 38]. In some situations, unconsciously processed stimuli could affect behavioral performance more profoundly than consciously processed stimuli [39]. For instance, a study found that unconsciously processed stimuli could enhance emotional memory after both short and long delays. This result indicated that emotion could enhance memory processing even when the stimuli was encoded unconsciously [40]. Another study found that people’s response to painful stimuli was more likely to be affected by unconscious negative emotions than the conscious negative emotions [14, 41]. Therefore, it’s imperative to study the processing of subliminal negative stimuli.”

d. In the sixth paragraph in your introduction, we should start by writing about what others have investigated, and then write something about what research is fully investigated and what we can do in the present study.

We really thank the Reviewer for this suggestion. We have revised it according to your advice.

Please see the sixth and seventh paragraph.

“In summary, the eye region was found to be the important area for conveying facial expression [25]. In previous research, the ERP research concerning empathy for pain mostly used pictures and videos describing hands or feet in painful situations, or they used a painful facial expression to explore the subjects’ index of brain activity and behavioral changes when observing others’ pain [26, 27, 28, 29, 30]. To date, there have been a few ERP studies comparing the effect of negative emotions on empathy for pain [26, 27, 31]. For example, one supraliminal ERP study looking at the effect of eye emotional information on empathy for pain found a significant difference on P3 component induced by different eye conditions. The amplitude in neutral eyes condition was greater than that in sad eyes condition and fear eyes condition. Different supraliminal emotional information expressed by eye region affects late processing of empathy for pain [31]. Further, negative emotions have a profound influence on human survival and are important for social adaption from an evolutionary perspective [32]. Additionally, subliminal stimuli can also have impact on human behavior [33, 34, 35, 36, 37, 38]. In some situations, unconsciously processed stimuli could affect behavioral performance more profoundly than consciously processed stimuli [39]. For instance, a study found that unconsciously processed stimuli could enhance emotional memory after both short and long delays. This result indicated that emotion could enhance memory processing even when the stimuli was encoded unconsciously [40]. Another study found that people’s response to painful stimuli was more likely to be affected by unconscious negative emotions than the conscious negative emotions [14, 41]. Therefore, it’s imperative to study the processing of subliminal negative stimuli.

In this study, we aimed to compare the neural response to subliminal sad and fearful negative emotions conveyed by eyes on empathy for pain with an ERP technique. We used a typically subliminal affective priming paradigm with an empathy task, which is similar to some previous studies [26, 27, 41], to investigate which negative emotion was highly correlated with empathy for pain automatically. The theory of emotion-sharing indicated that emotion-sharing between the self and others was the base of empathic behavior [42]. When individuals perceived others’ emotion, they automatically imitated others’ emotions and share the same representation. As we know, the fear emotion means danger and a crisis [43]. It can cause avoidance behaviors. In contrast, sadness can give rise to prosocial behaviors [44]. Furthermore, the priming pictures in one of the previous studies were situational pictures that stimulate participants’ own emotions [26]. While our procedure was different in term of that we used different eye emotions as cues to prime empathy for pain. The eye cues were external emotion sources. Hence, we could supply some evidence of finding the suitable emotion from others when empathy for pain occurred to explain the subsequent social behaviors such as helping others or escaping. ”

e. The two sentences “Then, how do humans perceive others’ pain in completely different situations? It is still unknown and needs to be further explored.” may be better expressed in a declarative sentence.

Thanks for the reviewer’s advice. We have expressed it in declarative sentence.

Please see the revised part of it:the seventh paragraph.

“In this study, we aimed to compare the neural response to subliminal sad and fearful negative emotions conveyed by eyes on empathy for pain with an ERP technique. We used a typically subliminal affective priming paradigm with an empathy task, which is similar to some previous studies [26, 27, 41], to investigate which negative emotion was highly correlated with empathy for pain automatically. The theory of emotion-sharing indicated that emotion-sharing between the self and others was the base of empathic behavior [42]. When individuals perceived others’ emotion, they automatically imitated others’ emotions and share the same representation. As we know, the fear emotion means danger and a crisis [43]. It can cause avoidance behaviors. In contrast, sadness can give rise to prosocial behaviors [44]. Furthermore, the priming pictures in one of the previous studies were situational pictures that stimulate participants’ own emotions [26]. While our procedure was different in term of that we used different eye emotions as cues to prime empathy for pain. The eye cues were external emotion sources. Hence, we could supply some evidence of finding the suitable emotion from others when empathy for pain occurred to explain the subsequent social behaviors such as helping others or escaping. ”

Methods

a. The authors write "Nineteen participants with no history of brain injury or mental disorder participated in the experiment." Then, we can see “Thirty participants” and “another 29 participants’ coming up in the Experimental Design and Stimuli part. Did the authors make some writing mistakes or some other reasons?

Thanks a lot for your question. Because we have two kinds of stimuli. One is the priming stimuli, the other is the target stimuli. As a result, we asked two groups to evaluate the stimuli. The thirty participants assessed the priming pictures. Another 29 participants assessed the target pictures. The nineteen new participants just took part in the ERP experiment. They have been highlighted in light blue in the new manuscript.

“Participants

Nineteen participants with no history of brain injury or mental disorder participated in the experiment. All participants had normal or corrected-to-normal vision, signed the informed consent form and received compensation after the experiment. Four of the participants’ data (no obvious classical ERP components for one of them, and valid trials in each condition were 51.8% for the other three) were rejected due to artifacts during electroencephalographic (EEG) recording. Data of 15 participants were retained (6 men, 9 women; 22.33 ± 1.91 years (M ± SD)). This study was approved by the Ethics Committee of the Academy of Psychology and Behavior, Tianjin Normal university.”

“Experimental Design and Stimuli

This study used an affective priming paradigm with a 3 priming type (subliminal neutral, sad, fear eye region information) × 2 target type (painful, nonpainful pictures) within-subject design. Therefore, there were 6 conditions in total: Neutral/ Painful, Neutral/ Non-Painful, Sad/ Painful, Sad/ Non-Painful, Fear/ Painful, Fear/ Non-Painful. The priming stimuli used in the experiment were 42 facial black and white pictures (comprising 14 subliminal sad, 14 subliminal fear, and 14 subliminal neutral pictures) selected from the Chinese Affective Face Picture System (CAFPS) [47]. As the total eye region from the CAFPS was not large enough, as well as that some eye region emotions were rated poorly (i.e. some sad eyes did not convey sad emotions sufficiently), a minority of them (7 pictures) were taken from the internet with only remaining the eye region. The gender of the characters in the images for each emotion type was equal. The targets were pictures showing a person’s hands in painful or nonpainful situations [48, 49]. The size of eye region pictures was adjusted to 260 pixels in width and 115 pixels in height. The size of masks was 255 × 105 pixels. They were obtained by randomly scrambling 10 × 10-pixel squares on every cropped face with MATLAB2016b. The size of targets was 369 × 287 pixels. The distance between the participants and the computer monitor was approximately 65 cm. The pictures were normalized for size, global contrast and luminance. 

Thirty participants assessed the priming pictures prior to the experiment (rated on a 9-point scale). For the level of arousal, both sad (M ± SD: 6.71 ± .56) and fearful eye regions (6.84 ± .38) differed significantly in priming from neutral eye regions (3.52 ± .20, p < .001), but the difference between the sad and fearful eye regions was not significant (p = .40). For level of valence, the sad (2.37 ± .43) and fearful eye regions (2.31 ± .73) differed dramatically from neutral eye regions (4.30 ± .56, p < .001), but the difference between the sad and fearful eye regions was not significant (p = .79). Another 29 participants assessed the target pictures (rated on a 4-point scale). The pain intensity of the painful pictures (M ± SD: 2.16 ± .22) and the nonpainful pictures (.30 ± .19) were significantly different (p < .001).”

b. What is “IRI-C scale”? I did not find the full name of this abbreviation, which should be indicated in the text. There exists other improper abbreviation, for example the “PD”.

Thanks for the Reviewer’s advice. The IRI-C scale we used is the Chinese version of the “Interpersonal Reactivity Index-Chinese”. The PD scale, a subscale of the IRI-C, is short for the Personal Distress. We have revised it in the new manuscript. 

We have given a brief description of IRI at the third paragraph of the “Experimental Procedure” section. Please see it in the new manuscript. This part has been highlighted in yellow in the new manuscript.

“After the EEG session, all participants were asked to complete the Chinese version of Interpersonal Reactivity Index scale (IRI-C scale) [3, 52], which contained 22 items. A 5-point Likert scale was used for responses, ranging from inappropriate (0 point) to very appropriate (4 points). 5 items were scored in reverse. In this questionnaire, empathy was divided into 4 parts: perspective-taking (tendency to spontaneously adopt the psychological point of view of others), empathy concern (“other-oriented” feelings of sympathy and concern for unfortunate others), fantasy (tendency to transpose themselves imaginatively into the feelings and actions of fictitious characters in books, movies, and plays) and personal distress (“self-oriented” feelings of personal anxiety and unease in tense interpersonal settings). Correlations between the personal distress subscale and ERP components was conducted in order to find out the relationship between state empathy in task and trait empathy [30]. ”

c. Why only make a correlation analysis between the pain effect and the Personal Distress (PD) subscales of the IRI-C? How about other scales of the IRI-C?

We referred to previous research about the correlation between the pain effect and the PD subscales of the IRI-C (Jiao, Wang, Peng, & Cui, 2017). And at the last paragraph of discussion, we have explained the reason in detail. What’s more, we also calculated the other scales of the IRI-C, but the correlations were not significant.

Results

The abbreviation “RT” is not clear. 

We have revised it.

“Behavioral Results

A three-way repeated measures ANOVA was conducted to examine the reaction time (RT) differences between the experimental conditions. The results are depicted in Fig 2.”

Discussion

a. The authors write “Fear indicates danger and a crisis, which can cause avoidance behaviors.” This finding might be caused by people being more affected by sad eye emotion and sad eye emotion information requiring more attention.” Is there any grammatical error here?

Thank you very much. We have asked the expert who were good at English to revise it. It has been highlighted in yellow in the new manuscript.The first paragraph of Discussion:

“Previous studies showed the significance of subliminal stimuli. For instance, subliminal stimuli that failed to reach awareness may affect performance more effectively than stimuli that do reach awareness [55]. Invisible stimuli that are irrelevant to task are more disturbing than visible distractors because of failed communication in regions of the brain that normally suppress such responses [55]. Meanwhile, a study found that the affectively significant visual stimuli had privileged access to consciousness [56]. Our research initially used subliminal eye region information as the experimental stimulus to explore its effect on empathy for pain. The eye region was found to be an important area for conveying facial expression [25]. The subliminal paradigm in the present study was used to highlight the automatic processing of eye cues. Furthermore, unlike previous studies that roughly divided the experimental materials into positive, neutral, and negative emotion types, the current research divided the types of negative emotions more elaborate. We chose neutral, sad and fearful emotions because sad and fearful emotions have great significance for human survival. Sadness can give rise to prosocial behaviors [44] and fear can cause avoidance behaviors. Therefore, in this study, we compared the neural response to subliminal sad and fearful negative emotions conveyed by eye cues on empathy for pain by measuring the ERP responses.”

b. The authors write “How do humans automatically perceive others’ pain in these different situations? We still need to explore this question”. This point may be not appreciated. I think it can be replaced by your goal and hypothesis of your research.

Thanks for your advice. We have replaced it with the goal of our research. It has been highlighted in yellow in the new manuscript.The fisrt paragraph of Discussion as above.

c. “Additionally, P2 is an index of attention [55]. It is enhanced with greater attention.” These two sentences have the same meaning, change the expression of a subordinate clause or delete one sentence.

Thanks a lot. We deleted it.

d. Psychology is very gender-biased. Whether gender influences the results in this study is worth further discussion. All participants have no history of brain injury or mental disorder in the study, does this phenomenon still exist in other groups or in other age groups?

We definitely appreciate the Reviewer’s suggestion. With the ERP technique, this study initially explored the effect of emotional information from eyes on empathy for pain. We discussed this question preliminarily. Therefore, the effect of gender difference in this phenomenon was not the major question. Similar to previous study (Fan & Han, 2008), what we studied was pervasive features of emotional information from eyes on empathy for pain. Besides, as we know, Han, Fan and Mao (2008) have studied the gender difference in empathy for pain. And we would explore whether this phenomenon in our study exists in different gender, age groups and even in some special subjects in future research.

Reviewer #2: This paper explored the effect of subliminal sad and fearful information conveyed by eyes on pain empathy by ERPs, and found that sad lead to slower response and greater P2 amplitude. The focus of the paper is interesting, the methods and used and the logic are reasonable, the structure of the article is clear. The following problems need further addressed.

Thank you so much for your suggestions.Please see the revised version of manuscript and manuscript with track changes. We also have Figure1,2,3,4(revised) version.

Major points:

1. This article should focus on the modulation of subliminal emotional primes on pain empathy, so authors need to pay more attention to the interactions between the primes and targets. For example, authors only report P3 and LPC are affected by painful / nonpainful targets, can subliminal emotion be modified this effect? The results and discussion are less concerned about interactions, the reported interactions in the results are also poorly explained.

We appreciate the Reviewer’s suggestion. But for the reason that we did not get interaction results between the primes and targets about P3 and LPC. Therefore, we did not explain it in the discussion.

On the other hand, we found an interaction between emotional primes and targets about the P2 component. In detail, the subliminal sad eye stimulus elicited a larger P2 amplitude than the subliminal fearful eye stimulus when participants were doing empathy for pain.

2. The author used the subliminal stimuli as primes, but in the introduction part, they did not present enough reasons about the significance of subliminal emotion and the difference between subliminal and supraliminal stimuli. The discussion part also lacked the related explanation.

We appreciate the reviewer’s suggestion and we fully agree that we should present enough reasons about the significance of subliminal emotion and the difference between subliminal and supraliminal stimuli. We have added these parts highlighted in yellow in both the “Introduction and discussion” parts. 

“Introduction

In summary, the eye region was found to be the important area for conveying facial expression [25]. In previous research, the ERP research concerning empathy for pain mostly used pictures and videos describing hands or feet in painful situations, or they used a painful facial expression to explore the subjects’ index of brain activity and behavioral changes when observing others’ pain [26, 27, 28, 29, 30]. To date, there have been a few ERP studies comparing the effect of negative emotions on empathy for pain [26, 27, 31]. For example, one supraliminal ERP study looking at the effect of eye emotional information on empathy for pain found a significant difference on P3 component induced by different eye conditions. The amplitude in neutral eyes condition was greater than that in sad eyes condition and fear eyes condition. Different supraliminal emotional information expressed by eye region affects late processing of empathy for pain [31]. Further, negative emotions have a profound influence on human survival and are important for social adaption from an evolutionary perspective [32]. Additionally, subliminal stimuli can also have impact on human behavior [33, 34, 35, 36, 37, 38]. In some situations, unconsciously processed stimuli could affect behavioral performance more profoundly than consciously processed stimuli [39]. For instance, a study found that unconsciously processed stimuli could enhance emotional memory after both short and long delays. This result indicated that emotion could enhance memory processing even when the stimuli was encoded unconsciously [40]. Another study found that people’s response to painful stimuli was more likely to be affected by unconscious negative emotions than the conscious negative emotions [14, 41]. Therefore, it’s imperative to study the processing of subliminal negative stimuli.

In this study, we aimed to compare the neural response to subliminal sad and fearful negative emotions conveyed by eyes on empathy for pain with an ERP technique. We used a typically subliminal affective priming paradigm with an empathy task, which is similar to some previous studies [26, 27, 41], to investigate which negative emotion was highly correlated with empathy for pain automatically. The theory of emotion-sharing indicated that emotion-sharing between the self and others was the base of empathic behavior [42]. When individuals perceived others’ emotion, they automatically imitated others’ emotions and share the same representation. As we know, the fear emotion means danger and a crisis [43]. It can cause avoidance behaviors. In contrast, sadness can give rise to prosocial behaviors [44]. Furthermore, the priming pictures in one of the previous studies were situational pictures that stimulate participants’ own emotions [26]. While our procedure was different in term of that we used different eye emotions as cues to prime empathy for pain. The eye cues were external emotion sources. Hence, we could supply some evidence of finding the suitable emotion from others when empathy for pain occurred to explain the subsequent social behaviors such as helping others or escaping. 

Discussion

Previous studies showed the significance of subliminal stimuli. For instance, subliminal stimuli that failed to reach awareness may affect performance more effectively than stimuli that do reach awareness [55]. Invisible stimuli that are irrelevant to task are more disturbing than visible distractors because of failed communication in regions of the brain that normally suppress such responses [55]. Meanwhile, a study found that the affectively significant visual stimuli had privileged access to consciousness [56]. Our research initially used subliminal eye region information as the experimental stimulus to explore its effect on empathy for pain. The eye region was found to be an important area for conveying facial expression [25]. The subliminal paradigm in the present study was used to highlight the automatic processing of eye cues. Furthermore, unlike previous studies that roughly divided the experimental materials into positive, neutral, and negative emotion types, the current research divided the types of negative emotions more elaborate. We chose neutral, sad and fearful emotions because sad and fearful emotions have great significance for human survival. Sadness can give rise to prosocial behaviors [44] and fear can cause avoidance behaviors. Therefore, in this study, we compared the neural response to subliminal sad and fearful negative emotions conveyed by eye cues on empathy for pain by measuring the ERP responses.

”

3. Methods and Results: the number of subjects (only 15 subjects’ results) was relatively small. In addition to arousal and valence, did the emotional types (sad, fearful or neutral?) be evaluated by subjects? Authors report interactions in the results, but do not report clearly how they interact in subsequent analyses from the view of two independent variables, so that readers can understand.

Thanks for your advice. Firstly, we have already assessed both the priming and target stimuli. We have written it in the second paragraph of the “Experimental Design and Stimuli” part.

Secondly, although we only have 15 valid participants, the results of our effect size (η2p) (Please check the value in the manuscript) was larger enough to indicate our results were stable. Moreover, the number of participants in some empathy research with ERP technique published also had less than 20 people (Jiao, Wang, Peng & Cui, 2017; Suzuki, Galli, Ikeda, Itakura & Kitazaki, 2015; Jiang, Varnum, Hou & Han, 2013; Mu, Fan, Mao & Han, 2008; Fan & Han, 2008). So, we think that our result was a sound one.

Thirdly, we also have reported interactions and the simple effect of the two independent variables’ interactions in subsequent analyses. Please see the detailed parts in the manuscript.

4. In view of the current results, there is a suggestion for data analysis. When analyzing ERP data, the author conducted repeated measurement analysis of brain region × primes (sad, fearful and neutral) × target type (painful or nonpainful), and the main results focused on the main effect of the region, while other results were not good enough. The ERP components differed in different brain regions, which is natural and easily understood results. Therefore, the analysis of specific ERP component can be conducted by analyzing the differences in brain regions first, and then analyze the electrodes of the most obvious region with prime types× target types measurements, which can more effectively reflect the author's intention.

Thanks a lot for your brilliant advice. The statistical method we used referred to previous studies (Cheng, Jiao, Luo, & Cui, 2017; Jiao, Wang, Peng, & Cui, 2017; Luo, Feng, He, Wang, & Luo, 2010; Mu & Han, 2013) and our own topographical distribution. So, we have such confidence to believe that our statistical method is authentic.

Especially for P2, the classical regions about empathy involved the frontal and central regions (Han, Luo & Han, 2015; Luo, Han, Du & Han, 2018). However, P2 involved the centro-parietal regions (Xu, Li, Ding, Zhang, Fan, Diao et al., 2015; Pesciarelli, Leo & Sarlo, 2016) and the parieto-occipital regions (Liu, Pinheiro, Zhao, Nestor, McCarley & Niznikiewicz, 2012) when the research was about emotion. In this study, P2 was calculated via the frontal, frontal-central, central, centro-parietal and parietal regions on account of this study contained empathy and emotion (Cheng, Jiao, Luo & Cui, 2017). Because of the uncertainty in the studies about P2 until now, we chose wide regions to study the feature for P2 in empathy field, especially in our subliminal design.

We did consider your advice thoroughly. Our work was the first one to detect the eye region effect on pain empathy with subliminal paradigm which might have less effect than that in supraliminal paradigm, so we wanted to study the difference through wide brain regions to find out the changing tendency in our experimental design in order not to miss some vital findings. And it might be provide some evidence for further study which might be focused on special region for special effect. It did have some creative value in empathy field. And we really did the analysis on the basis of a series of references in this field. The region we chose was from frontal to parietal, which can make our analysis more consistent among different components. We just followed the main clue. If there was interaction, we reported it. If there was none, we did not further analyse it. 

If you have more opinions, we really appreciate you can instruct us, discuss with us and please give us another chance to revise it. Thank you so much again.

Minor points:

Figure 1: It is suggested to replace the eyes, mask and target of the actual material samples.

Thanks for your suggestion. We have revised it.

Please see the Figure1 (revised).

Figure 2: To add SE as error bar to the bar of reaction time. The unit and description should be added on the axis. The vertical axis should start from 0, and statistics significance is suggested to be added.

Really thanks for your detailed suggestion. We have repainted the pictures. It looks nice now.

Please see the Figure2 (revised).

Figure 3: The ERP figure is not clear enough to see the detail including time axis. There are six lines in the ERP figure. It is recommended to take the peaks and draw a bar chart to express the results more clearly.

Thanks a lot. We have repainted according to your advice.

Please see the Figure3 (revised).

References

Cheng, J., Jiao, C., Luo, Y., & Cui, F. (2017). Music induced happy mood suppresses the neural responses to other's pain: Evidences from an ERP study. Scientific Reports, 7(1).

Fan, Y., & Han, S. (2008). Temporal dynamic of neural mechanisms involved in empathy for pain: An event-related brain potential study. Neuropsychologia, 46(1),160-173.

Han, S., Fan, Y., & Mao, L. (2008). Gender difference in empathy for pain: An electrophysiological investigation. Brain Research, 1196, 85-93.

Han, X. C., Luo, S. Y., & Han, S. H. (2015). Embodied neural responses to others’ suffering. Cognitive Neuroscience, 7(1–4), 114–127.

Jiang, C., Varnum, M. E. W., Hou, Y., & Han, S. (2013). Distinct effects of self-construal priming on empathic neural responses in Chinese and Westerners. Social Neuroscience, 9(2),130-138.

Jiao, C., Wang, T., Peng, X., & Cui, F. (2017). Impaired empathy processing in individuals with internet addiction disorder: An Event-Related Potential study. Frontiers in Human Neuroscience, 11, 498. 

Liu, T., Pinheiro, A., Zhao, Z., Nestor, P., McCarley, R., & Niznikiewicz, M. (2012). Emotional cues during simultaneous face and voice processing: Electrophysiological insights. PLoS ONE, 7(2), e31001.

Luo, W., Feng, W., He, W., Wang, N., & Luo, Y. (2010). Three stages of facial expression processing: ERP study with rapid serial visual presentation. NeuroImage, 49(2), 1857-1867.

Luo, S., Han, X., Du, N., & Han, S. (2018). Physical coldness enhances racial in-group bias in empathy: Electrophysiological evidence. Neuropsychologia, 116, 117-125.

Mu, Y., Fan, Y., Mao, L., & Han, S. (2008). Event-related theta and alpha oscillations mediate empathy for pain. Brain Research, 1234, 128-136.

Mu, Y., Han, S. (2013). Neural oscillations dissociate between self-related attentional orientation versus evaluation. NeuroImage, 67, 247-256.

Parkington K.B., Itier R. J. (2018). One versus two eyes makes a difference! Early face perception is modulated by featural fixation and feature context, CORTEX.

Pesciarelli, F., Leo, I., & Sarlo, M. (2016). Implicit Processing of the Eyes and Mouth: Evidence from Human Electrophysiology. PLOS ONE, 11(1), e0147415.

Suzuki, Y., Galli, L., Ikeda, A., Itakura, S., & Kitazaki, M. (2015). Measuring empathy for human and robot hand pain using electroencephalography. Scientific Reports, 5(1).

Xu, M., Li, Z., Ding, C., Zhang, J., Fan, L., Diao, L., & Yang, D. (2015). The divergent effects of fear and disgust on inhibitory control: An ERP study. PLoS ONE, 10(6), e0128932.

---

## [Editor Report · Decision Letter 1]

22 Nov 2019

The effect of emotional information from eyes on empathy for pain: A subliminal ERP Study

PONE-D-19-16381R1

Dear Dr. Song,

We are pleased to inform you that your manuscript has been judged scientifically suitable for publication and will be formally accepted for publication once it complies with all outstanding technical requirements.

With kind regards,

Kai Wang

Academic Editor

PLOS ONE
---

## [Editor Report · Acceptance letter]

6 Dec 2019

PONE-D-19-16381R1 

The effect of emotional information from eyes on empathy for pain: A subliminal ERP Study 

Dear Dr. Song:

I am pleased to inform you that your manuscript has been deemed suitable for publication in PLOS ONE. Congratulations! Your manuscript is now with our production department. 

With kind regards,

on behalf of

Prof. Kai Wang 

Academic Editor

PLOS ONE